# Characterization of the Heavy-Metal-Associated Isoprenylated Plant Protein (*HIPP*) Gene Family from *Triticeae* Species

**DOI:** 10.3390/ijms21176191

**Published:** 2020-08-27

**Authors:** Heng Zhang, Xu Zhang, Jia Liu, Ying Niu, Yiming Chen, Yongli Hao, Jia Zhao, Li Sun, Haiyan Wang, Jin Xiao, Xiue Wang

**Affiliations:** 1State Key Lab of Crop Genetics and Germplasm Enhancement, Cytogenetics Institute, Nanjing Agricultural University/JCIC-MCP, Nanjing 210095, China; 2016201031@njau.edu.cn (H.Z.); 2018201061@njau.edu.cn (X.Z.); 2018201063@njau.edu.cn (J.L.); 2017101125@njau.edu.cn (Y.N.); 2019101119@njau.edu.cn (Y.C.); haoyongli9128@163.com (Y.H.); jiazhao@scau.edu.cn (J.Z.); sunli@njau.edu.cn (L.S.); hywang@njau.edu.cn (H.W.); 2College of Agriculture, South China Agriculture University, Guangzhou 510642, China

**Keywords:** HIPP, gene family, *Haynaldia villosa* L., subcellular localization, Cd tolerance

## Abstract

Heavy-metal-associated (HMA) isoprenylated plant proteins (HIPPs) only exist in vascular plants. They play important roles in responses to biotic/abiotic stresses, heavy-metal homeostasis, and detoxification. However, research on the distribution, diversification, and function of *HIPPs* in *Triticeae* species is limited. In this study, a total of 278 *HIPPs* were identified from a database from five *Triticeae* species, and 13 were cloned from *Haynaldia villosa*. These genes were classified into five groups by phylogenetic analysis. Most HIPPs had one HMA domain, while 51 from Clade I had two, and all *HIPPs* had good collinear relationships between species or subgenomes. In silico expression profiling revealed that 44 of the 114 wheat *HIPPs* were dominantly expressed in roots, 43 were upregulated under biotic stresses, and 29 were upregulated upon drought or heat treatment. Subcellular localization analysis of the cloned *HIPPs* from *H. villosa* showed that they were expressed on the plasma membrane. *HIPP1-V* was upregulated in *H. villosa* after Cd treatment, and transgenic wheat plants overexpressing *HIPP1-V* showed enhanced Cd tolerance, as shown by the recovery of seed-germination and root-growth inhibition by supplementary Cd. This research provides a genome-wide overview of the *Triticeae HIPP* genes and proved that *HIPP1-V* positively regulates Cd tolerance in common wheat.

## 1. Introduction

In recent decades, heavy-metal contamination in the environment has increased due to the rapid development of industry and the use of pesticides [1,2]. In some parts of the world, such as Ziyang County, China [3], soil conditions are detrimental to the growth of many or most plant species due to the increasing level of certain inorganic ions [4]. Sometimes, even low concentrations of heavy metals such as Cd, Pb, Al, and Hg can cause great damage to plants [5,6,7,8]. Heavy metals generally produce toxic effects in plants, including chlorosis, inhibition of growth and photosynthesis, low biomass accumulation, imbalance of nutrient assimilation and water, and senescence, which eventually lead to plant death [9]. Cd is a nonessential heavy metal and is ranked among the top 20 toxins [10], and it can affect many processes of plant growth and development. Contamination of soil with 5 μM Cd can significantly affect the seed germination and seedling growth of barley [11]. Different concentrations of Cd can also affect the root growth of wheat and rice [12,13]. Photosynthetic indices of tomato seedlings, such as the photosynthetic rate (Pn) and the intracellular CO_2_ concentration (Ci), can be severely affected under Cd stress [14]. In order to survive in environments containing heavy metals, plants have to develop a range of strategies to cope with these heavy metals [9]. Research reports that several protein families are involved in the detoxification and sequestration of heavy metals, such as ATP-binding cassette (ABC) transporters [15], zinc/iron-regulated transporter proteins (ZIPs) [16], heavy-metal ATPases (HMAs) [17,18,19], cation diffusion facilitators (CDFs) [20], natural resistance-associated macrophage proteins (Nramps) [21], and heavy-metal-associated isoprenylated plant proteins (HIPPs) [22].

HIPPs are a group of metal-binding metallochaperones characterized by a heavy-metal-associated (HMA) domain and an isoprenylation motif [23]. Although the HMA domain and the isoprenylation motif commonly occur in many organisms, from bacteria to humans, the presence of both interacting in the same protein has been observed only in vascular plants [24]. Analysis of *HIPP-*family genes has mainly been done for *Oryza* and *Arabidopsis*, with a few studies in other species [23,24]. Thus far, 45 *HIPP* genes have been identified in *Arabidopsis*, 59 in *Oryza*, 74 in *Populus trichocarpa*, 52 in *Setaria italic*, and 5 in *Selaginella moellendorffii*, and these genes have been divided into five distinct clusters [23]. All HIPP proteins have a conserved structure, including an HMA domain and a C-terminal isoprenylation CaaX motif (where “C” is cysteine, “a” is an aliphatic amino acid, and “X” is any amino acid); some of them also contain other domains, such as glycine-rich repetitions and a proline-rich motif [23].

The functions of *HIPPs* have been extensively studied in *Arabidopsis* [25], *Oryza* [26], tomato [27], barley [28], wheat [29], etc., and *HIPPs* have been revealed to play an important role in the maintenance of heavy-metal homeostasis and detoxification. In *Arabidopsis*, the expression of *AtCdI19* can be induced by Cd, Hg, Fe, and Cu, and overexpression of *CdI19* confers Cd tolerance in transgenic *Arabidopsis* [30]. Expression of *Arabidopsis HIPP20*, *HIPP22*, *HIPP26*, and *HIPP27* in yeast confers increased Cd resistance to the Cd-sensitive yeast strain *ycf1*. The *hipp20/21/22* triple mutant is more sensitive to Cd and shows significantly decreased shoot fresh weight compared to the wild-type [24]. *AtFP6* expression can be induced by Cd and Zn and the protein can bind Pb, Cd, and Cu; overexpression of *AtFP6* can enhance tolerance to Cd compared to the wild-type [22]. AtHIPP44 can interact with the transcription factor MYB49, thus leading to its upregulated expression and the subsequent reduction of Cd accumulation [31]. In rice, the expression of *OsHIPP16*, *OsHIPP28*, *OsHIPP34*, *OsATX1*, and *OsHIPP60* in roots and shoots can be induced by Mn, Cd, and Cu [26]. The expression of *OsHIPP34*, *OsHIPP60*, and *OsHIPP16* in a yeast mutant showed that *OsHIPP34* can increase resistance to Cu, *OsHIPP60* can increase resistance to Zn, and *OsHIPP16* can increase resistance to Cd and Zn [26]. Additionally, the *oshipp42* mutant grows more weakly than the wild-type under Cu, Zn, Cd, and Mn stresses [26]. *OsHIPP29* is upregulated by high Cd and Zn concentrations in the shoots and roots; the mutants and RNAi lines of *OsHIPP29* show decreased plant heights and dry biomass compared to the wild-type under Cd exposure [32]. Some *HIPP* genes may also be involved in other abiotic-stress responses. *HvFP1* from barley shows a complex expression pattern with induction under different abiotic stress conditions (e.g., cold, drought, and heavy-metal exposure) during leaf senescence and in response to abscisic acid [28]. Like *HvFP1*, *HIPP26* from *Arabidopsis* can also be induced by cold, salt, and drought stresses [33]. Evidence has accumulated for the critical role of HIPP in response to biotic stresses. In *Arabidopsis*, AtHIPP3 acts as an upstream regulator of the salicylate-dependent pathway of pathogen response [25]. HIPP27 is a host-susceptibility factor required for beet cyst nematode infection and development [34]. In tobacco, NbHIPP26 interacts with TGB1 (the potato mop-top virus movement protein) to activate the drought-stress response and to facilitate the long-distance movement of the virus [35]. In wheat, transient silencing of *TaHIPP1* enhances stripe-rust resistance, indicating that *TaHIPP1* acts as a negative regulator [36].

Common wheat (*Triticum aestivum* L.) is one of the most important food crops around the world, and the growth and yield of wheat are affected by various environmental factors, such as heavy metals [37]. To date, some genes related to heavy metal tolerance have been reported in wheat, including *TaVP1* [38], *TaGolS3* [37], *TaEXPA2* [39], *TaHMA2* [40], *TaPCS1* [41], *TaCNR2* [42], and *TaPCS1* [43]. In this study, genome-wide identification of the *HIPP* gene family in *Triticeae* species was performed by searching the published sequences. A phylogenetic tree of the *HIPP* genes was constructed and the evolutionary relationships were analyzed. Chromosome distribution and protein structure were further studied to gain a better understanding of *HIPP* genes in wheat. The potential functions of the common wheat *HIPP* genes were predicted through their expression profiles based on in silico analysis. *Haynaldia villosa* L. (2*n* = 2*x* = 14, VV) is a diploid wild relative of wheat. Previous studies have shown that *H. villosa* is resistant to various wheat diseases such as stripe rust and powdery mildew, and also possesses the characteristics of resistance to drought, cold, and salt; therefore, it provides excellent material for wheat genetic improvement [44,45,46]. As *H. villosa* has not been sequenced, we cloned *H. villosa*’s *HIPPs* through homology-based cloning. One of the *HIPP* genes, *HIPP1-V*, was transformed into wheat to study its role in Cd tolerance. Our results will support the understanding of the evolution and diversification of *HIPPs* in *Triticeae* species at a genome-wide scale.

## 2. Results

### 2.1. Genome-Wide Identification, Phylogenetic Analysis, and Protein Structure Analysis of the HIPP Gene Family in Triticeae Species

In total, 278 *HIPP* genes from five *Triticeae* species were identified from the public database. Of these, 114 were from common wheat (*T. aestivum*), 33 from *Triticum urartu*, 40 from *Aegilops tauschii*, 58 from *Triticum dicoccoides*, and 33 from *Hordeum vulgare*. By using the homology cloning technique, 13 *HIPPs* from *H. villosa* were cloned (Figure 1). To understand the phylogeny of *HIPP* genes, the evolutionary relationships of the above 291 *Triticeae HIPPs* (278 from five *Triticeae* species and 13 from *H. villosa*), along with 40 from *Brachypodium distachyon*, 59 from rice, and 45 from *Arabidopsis*, were phylogenetically analyzed and a neighbor-joining tree was built (Figure 2 and Appendix A). The *HIPPs* were divided into five clades (Clades I–V) according to the classification established for *Arabidopsis*. Clade IV was the smallest clade with 11 members, while Clade II was the largest with 128 members. Apart from *H. villosa*, the other analyzed *Triticeae* species possessed genes in each of these clusters. In the same clade, *HIPP* genes from different *Triticeae* species were clustered together, and *HIPP* genes from the same species also tended to aggregate (Figure 2), showing differences between the *HIPP* genes of different species and the conservation of *HIPP* genes within the same species.

The conserved domains of the HIPP proteins were evaluated using the Conserved Domain Database. All HIPP proteins contained 1 or 2 HMA domains (pfam00403) and a C-terminal isoprenylation CaaX motif (Appendix A). Significant differences in the genes’ functional domains were observed between different clades (Appendix A); for example, the 51 HIPPs in Clade I had two HMA domains and the others all had one. Members of the same clade resembled one another, e.g., the eight genes in Clade IV were similar in size, ranging from 139 to 175 amino acids (aa), and they all had an HMA domain and an isoprenylation motif (Appendix A). Minor differences within the same clades were observed, e.g., the sizes and functional domains of Clade V were more diversified; the coding sequence (CDS) lengths of the members of this clade ranged from 77 to 1089 aa and some genes also contained other functional domains, e.g., a copA superfamily domain was found in TaHIPP7-A, TdHIPP7-A, and TuHIPP7; TaHIPP46-A contained an NB-ARC superfamily domain (Appendix A), etc.

### 2.2. Chromosomal Distribution of HIPPs

Based on the genome databases data for common wheat, *T. dicoccoides*, *T. urartu*, *Ae. tauschii*, and *H. vulgare*, the chromosome constitution of the *HIPP* family was investigated (Figure 3 and Table 1). The *HIPPs* were distributed on all of the chromosomes of the diploid genomes and polyploid subgenomes, indicating that the *HIPP* genes were distributed on the chromosomes of each homologous group. However, the number of *HIPP* genes of different homeologous groups varied greatly, with a minimum of 1–3 (the sixth and seventh homoeologous groups) and a maximum of 8–11 (the second homoeologous group) (Table 1). For homoeologous chromosomes from different genomes of wheat and its ancestral/related species, most of the corresponding *HIPP* orthologs were present in the syntenic genome regions. For example, *HIPP1*, *HIPP7*, *HIPP16*, *HIPP20*, *HIPP27*, and *HIPP29* were on the homoeologous Group 2 chromosomes in *Triticeae* species, while *HIPP37* and *HIPP39* were on the Group 3 chromosomes (Figure 3a–d). However, there were some exceptions; for example, *HIPP4* and *HIPP6* were present on Chromosome 2D, but not on Chromosome 2A or 2D in *T. aestivum* (Figure 3a). Furthermore, *HIPP21*, *HIPP30*, and *HIPP12* were on 4AL of *T. aestivum* and *T. dicoccoides* (Figure 3a,b), but on 4AS of *T. urartu* (Figure 3d), indicating that Chromosome 4A of tetraploid and hexaploid experienced structural rearrangement during its evolution. The *HIPP* genes of the other chromosomes were potential orthologs that showed a good collinearity (Figure 3).

Comparing the A genomes of common wheat, *T. dicoccoides*, and *T. urartu*, four *HIPPs* were found on Chromosome 3A and three on Chromosome 6A in all three species (Table 1). However, there were differences on the other chromosomes, especially in the number of *HIPPs* on Chromosome 1A—five in common wheat, two in *T. dicoccoides*, and two in *T. urartu*. One possible reason for this is that *HIPP* has replicated and formed gene clusters over the course of the evolution of common wheat (Table 1). Moreover, the number of *HIPP* genes found on Chromosome 4A was also very different between the three species, with seven in common wheat, five in *T. dicoccoides*, and four in *T. urartu* (Table 1). Comparing the B subgenomes of common wheat and *T. dicoccoides*, it was found that except for Chromosome 7B (three *HIPPs*), the number of *HIPPs* was different on the other chromosomes, of which 1B had the largest difference, with four in common wheat and one in *T. dicoccoides*, and the number of *HIPP* genes on the remaining chromosomes differed by one (Table 1). Comparing the D genomes of common wheat and *Ae. tauschii*, the total number of *HIPP* genes was 42 and 40, respectively. The numbers of genes on Chromosomes 1D, 6D, and 7D were the same, with five, three, and two, respectively. There was one more *HIPP* gene on Chromosomes 2D, 3D, and 5D in common wheat than in *Ae. tauschii*, and one less *HIPP* gene on Chromosome 4D in common wheat than in *Ae. tauschii*. This shows that the number of *HIPP* genes in the D genome has been relatively stable over the course of its evolution (Table 1).

### 2.3. The Expression Pattern of the HIPP Gene Family in Common Wheat

To predict the potential functions of the identified *HIPPs*, we investigated their expression in different tissues or in responses to various biotic and abiotic stresses through in silico expression profiling. The expression patterns of wheat *HIPPs* in different tissues (e.g., roots, leaves, spikes, and grains) under two biotic stresses (e.g., stripe-rust pathogen CYR31 and powdery-mildew pathogen E09) and two abiotic stresses (e.g., drought and heat) were investigated using the wheat RNA-seq data from publicly available databases. The expression level was measured as tags per million (TPM), and the Z-score-normalized values of the TPM were used to generate a heat map in R Studio (Figure 4). Tissue-specific expression analysis showed that 44 genes had the highest expression levels in the roots (e.g., *TaHIPP43-A*), 23 genes in the leaves (e.g., *TaHIPP2-A*), 42 genes in the spikes (e.g., *TaHIPP29-B*), and only *TaHIPP20-A*, *TaHIPP20-B*, *TaHIPP21-A*, *TaHIPP21-D*, and *TaHIPP36-B* in the grains (Figure 4a). Most *HIPP* genes showed the highest expression levels in the roots and the lowest in the grains, and the numbers of *HIPP* genes highly expressed in the spikes and leaves were somewhere in between. Different genes showed various expression patterns under biotic (e.g., *Bgt* and *Pst*) and abiotic (e.g., drought and heat) stresses. In response to *Bgt* and *Pst* infection, 52 and 69 genes were significantly upregulated, 31 and 13 genes were significantly downregulated, and 31 and 32 genes remained unchanged, respectively (Figure 4b,c). In response to drought and heat induction, 44 and 40 genes were significantly upregulated, 22 and 31 genes were significantly downregulated, and 48 and 43 genes remained unchanged, respectively (Figure 4d,e). When infected by *Bgt* and *Pst*, 41 genes (e.g., *TaHIPP13-B*) were upregulated (Appendix A), six genes (e.g., *TaHIPP36-D*) were downregulated (Appendix A), and 28 genes (e.g., *TaHIPP36-B*) showed no obvious expression change (Appendix A). Under drought and heat stresses, 29 genes (e.g., *TaHIPP35-D*) were upregulated (Appendix A), 15 genes (e.g., *TaHIPP24-B*) were downregulated (Appendix A), and 40 genes (e.g., *TaHIPP10-A*) showed no obvious expression change (Appendix A). When combining the four kinds of stresses, 14 genes (e.g., *TaHIPP35-A*) were upregulated (Appendix A) and none were downregulated (Appendix A).

### 2.4. Cloning and Subcellular Localization Analysis of the HIPPs from H. villosa

The 13 cloned *HIPPs* from *H. villosa* were assigned the names *HIPP1-V* to *HIPP13-V*, which were classified into four clades. *HIPP7-V*, which encodes a protein of 151 aa, had the shortest CDS length, e.g., 456 bp. The longest one was *HIPP2-V* at 1251 bp, which encodes 416 aa. The predicted isoelectric points of the HIPPs-V ranged from 5.21 (HIPP13-V) to 10.04 (HIPP10-V) (Table 2).

Investigating the subcellular localization of a protein provides clues towards the elucidation of its function. To verify where the HIPPs-V proteins were localized in *vivo*, GFP (green fluorescent protein)-tagged fusion proteins were expressed and observed in the leaves of *Nicotiana Benthamiana* via transient expression using the *Agrobacterium* method. The GFP signal of the fusion proteins could be detected for all 13 cloned HIPP-V proteins, with different localization signals (Figure 5). Compared to the control, which had an even distribution of GFP fluorescence (Figure 5a), all HIPP-V proteins were localized on the plasma membrane (PM), but with different fluorescence intensities, e.g., the fluorescence intensities of HIPP2, -4, -5, -7, -9, and -11-V were weaker than those of the others (Figure 5c,e,f,h,j,l). Some HIPPs also had obvious signals in the cytoplasm, such as HIPP2-V and HIPP9-V (Figure 5c,j). These different subcellular localizations revealed that they may be involved in multiple biological processes and may have distinct functions.

### 2.5. HIPP1-V Positively Regulates Cd Tolerance in Common Wheat

Previous studies have demonstrated the involvement of *HIPPs* in cadmium (Cd) homeostasis. Quantitative reverse-transcription PCR (qRT-PCR) analysis was used to investigate *HIPP1-V* expression in response to Cd. Following the treatment of *H. villosa* at the two-leaf stage with 1 mM Cd^2+^, *HIPP1-V* expression in the roots rapidly increased 1 h after inoculation (hai), reached a peak level at 12 hai, and remained at a higher level from 24 hai (Figure 6a). In the stems, *HIPP1-V* expression increased at 2 hai, declined at 6 hai, increased at 12 hai, and remained at a high level for all of the tested time points (Figure 6b). In the leaves, *HIPP1-V* expression increased at 2 hai, reached a peak level at 12 hai, and declined at 24 hai (Figure 6c). These results showed that *HIPP1-V* was upregulated after Cd induction. Although the expression trend was different in different tissues of *H. villosa*, they were all upregulated.

The function of *HIPP1-V* in Cd tolerance was validated using stable transformation. The expression vector pBI220: HIPP1-V was constructed by cloning the *HIPP1-V* gene into pBI220 driven by the 2 × 35S promoter. The vectors pBI220: HIPP1-V and pAHC20 (carrying a Bar gene as a selection marker) were co-transformed into Yangmai158 via particle bombardment, and five positive transgenic plants were obtained (Appendix A). OE-HIPP1-T_0_-17-3 and OE-HIPP1-T_0_-27-3 were selected for further analysis due to their higher *HIPP1-V* expression levels (Appendix A). Their derived T_3_ lines, OE-HIPP1-T_3_-1 and OE-HIPP1-T_3_-2, were identified as positive lines and used for Cd treatment. When treated with 0 µM Cd, there were no differences in plant height and root length between receptor Yangmai158 and OE-HIPP1-T_3_ (Figure 6d–f). As the concentration of Cd increased, the root length and plant height of both plants gradually decreased, but the decrease was more pronounced in Yangmai158 plants (Figure 6d,e). When the concentration of Cd reached 100 µM, the differences in root length and plant height were the greatest; Yangmai158 and OE-HIPP1-V-T_3_ were 8.19 cm and 11.06 cm in height (Figure 6d,g) and 3.03 cm and 5.75 cm in root length, respectively (Figure 6e,g). As the concentration of Cd continued to increase, the growth status of Yangmai158 and OE-HIPP1-V-T_3_ tended to be consistent (Figure 6d,e). When the concentrations of Cd reached 10 mM and 20 mM, the seeds of OE-HIPP1-V-T_3_ were still able to germinate, but the germination of Yangmai158 was completely suppressed (Figure 6h,i). These results indicate that *HIPP1-V* could significantly improve the tolerance of common wheat to Cd.

## 3. Discussion

### 3.1. The Cd Tolerance of HIPP1-V Could Expand Wheat Breeding Resources

Wheat is an important crop worldwide and is known to be a leading dietary source of Cd [47]. Improving wheat tolerance to Cd and reducing Cd accumulation in grains are issues that need to be addressed in current wheat breeding. Some Cd-tolerance-related genes in wheat have been reported; including HMA-containing genes [29]. This HMA domain has a highly conserved CysXXCys (X refers to any amino acid) motif comprising a beta–alpha–beta–alpha fold shape for binding heavy metals, such as Cd, Cu, or Zn [48]. There have been two types of HMA-domain-containing proteins found in plants to date, namely HPPs (HMA plant proteins) and HIPPs (HMA isoprenylated plant proteins), which are a group of metal-binding metallochaperones [23,26]. These proteins play crucial roles in metal homeostasis and detoxification in plants. The wheat heavy-metal ATPase 2 (TaHMA2) can transport Zn^2+^ and Cd^2+^ across membranes, and overexpression of *TaHMA2* in *Arabidopsis* increases the root length and fresh weight and enhances Zn^2+^/Cd^2+^ root-to-shoot translocation compared to that of the wild-type [40,49]. TaHIPP1 from common wheat contains an HMA domain, and overexpression of *TaHIPP1* in yeast significantly increases the cell growth rate under Cu^2+^ stress [29]. The cell number regulator 2 (TaCNR2) from common wheat is also involved in regulating heavy-metal translocation; the expression of *TaCNR2* in wheat seedlings increases under Cd, Zn, and Mn treatments, and overexpression of *TaCNR2* in *Arabidopsis* and rice plants enhances their tolerances to Cd, Zn, and Mn stresses. Using new gene resources from alien species is a very useful approach for breeding new cultivars [50]. In this study, *HIPP1-V,* which contains an HMA domain, was upregulated in the roots, stems, and leaves of *H. villosa*, induced by Cd. The overexpression of *HIPP1-V* was able to increase the tolerance of common wheat to Cd. This result indicates that *HIPP1-V* may represent a gene resource for improving wheat tolerance to Cd stress.

### 3.2. Identification and Characterization of the HIPP Gene Family Can Help to Better Understand the Evolutionary Relationships in Triticeae Species

HIPP proteins contain an HMA domain and a conserved isoprenylated motif, and are found only in vascular plants [23,51]. *HIPP*-family genes have been reported mostly in *Arabidopsis* and rice [23,26]. de Abreu-Neto speculated that plant metallochaperone genes (including *ATX*, *HPP*, and *HIPP*) share a common ancestor in plants and algae species, and started diversification early in plant evolution [23]. The distribution of the *HIPP* genes on chromosomes and segmental duplication events highlight the *HIPP* family’s expansion in the rice genome [26]; such gene duplication during evolution can help plants adapt to various environmental stresses [52]. In this study, a total of 291 *HIPP* genes were identified from five *Triticeae* species and *H. villosa*, and the *HIPP* gene family was divided into five clades; apart from *H. villosa*, the other *Triticeae* species possessed genes in each of these clusters. In *Triticeae* species, the number of *HIPPs* increased in fold accordingly in different ploidy species—the diploids *T. urartu*, *Ae. tauschii*, and *H. vulgare* had 33, 40 and 33, respectively, while the tetraploid *T. dicoccoides* had 58 and the hexaploid *T. aestivum* had 114. Chromosome location analysis indicated that the *HIPP* genes were distributed on the chromosomes of each homologous group, while protein structure analysis showed that the structures of the HIPP proteins in the same clade were similar. This indicates that most *HIPP* genes were formed before the differentiation of *Triticeae* species. Only a few genes might have evolved independently after the differentiation of *Triticeae* species, or this result might be due to the lack of genomic sequence information.

### 3.3. The HIPP Gene Family Might Regulate Plant Development and Various Responses to Stress in Common Wheat

Roles of HIPP proteins in heavy-metal homeostasis and/or detoxification [23], cold- and salt-stress responses [28], virus long-distance movement [35], plant–pathogen interactions, and plant development [25] have been demonstrated. *HIPP20*, *HIPP22*, *HIPP26*, *HIPP27*, *AtFP6*, *AtCdI9*, and *ATX1* from *Arabidopsis* and *OsHIPP16*, *OsHIPP28*, *OsHIPP34*, *OsATX1*, *OsHIPP60*, and *OsHIPP29* from rice can positively regulate tolerance to heavy metals [22,23,24,26,30,32,53]. Different genes can be induced by different heavy metals, which, in turn, increase plant tolerance to different heavy metals. *AtCdI9* can be induced by Cd, Hg, Fe, and Cu [30], while *OsATX1* and *OsHIPP60* can be induced by Mn, Cd, and Cu [26]. *OsHIPP34*-expressing yeast cells grow better in medium containing 60 μM Cu, but show no difference in medium containing Mn, Cd, and Zn compared to the control. Meanwhile, *OsHIPP60-*expressing yeast cells grow better under Cd and Zn treatment, but show no difference under treatment with Mn and Cu compared to the control [26]. Some HIPP proteins can also bind to specific heavy metals: *ATFP3* binds to transition-metal ions (e.g., Cu, Zn, and Ni) in a reversible manner [51], HIPP3 from *Arabidopsis* can combine with Zn [25], and AtFP6 can combine with some divalent cations, such as Pb^2+^, Cd^2+^, and Cu^2+^. The expression profiles of *HvFP1* and *HIPP26* are induced by cold and drought [28,33]. *NbHIPP26* is expressed specifically in the vascular parenchyma and is induced by drought and PMTV (Potato Mop-Top Virus) infection—when *NbHIPP26* is knocked down, the long-distance movement of the virus is suppressed [35]. HIPP3 acts as an upstream regulator of the salicylate-dependent pathway of pathogen response, and is also involved in abiotic-stress responses and seed and flower development [25]. Gene expression profiling analysis may provide important clues to gene functions. *HIPP1-V* from *H. villosa* was upregulated under Cd treatment, and overexpression of *HIPP1-V* indeed improved wheat tolerance to Cd stress. In this study, the *HIPP* genes showed different expression patterns in different tissues and different tolerances in wheat. *TaHIPP16-B* and *TaHIPP7-B* were highly expressed in the roots and were only upregulated in response to *Bgt* inoculation; *TaHIPP24-D* was highly expressed in the roots and was upregulated in response to *Bgt* inoculation and *Pst* infection; *TaHIPP29-B* was highly expressed in the spikes and was upregulated in response to drought and heat treatments. These results reveal the involvement of the different *HIPPs* in different biological processes; however, their specific roles need to be investigated by intensive functional analyses, such as the transgenic approach.

### 3.4. Diversification of the Subcellular Localization Pattern of HIPP in H. villosa

Protein localization and functions are related [54]. Isoprenylation, also known as farnesylation, is a post-translational protein modification that involves the addition of a C-terminal hydrophobic anchor, which is important for the interaction of a protein with membranes or other proteins [23,55]. All HIPP proteins contain an isoprenylation motif, which is then modified by isoprenylation. To the best of our knowledge, except for HIPP3 from *Arabidopsis*, all reported HIPP proteins localized on PM, including AtFP6 from *Arabidopsis* [22], NbHIPP26 from *N. Benthamiana* [35] and OsHIPP29 from rice [32]. Mutation in the isoprenylation domain may cause changes in protein localization; for instance, AtIPT3 farnesylation directs its localization in the nucleus/cytoplasm, whereas the nonfarnesylated protein is located in the plastids [56]. The wild-type CdI19 (GFP-CdI19W) is localized on the PM, whereas the farnesylated motif mutant CdI19 (in which the Cys-389 residue is replaced with Gly, GFPCdI19M) is localized in the entire cytoplasm, as is GFP [30]. Protein farnesylation of HIPPs has direct impacts on their function. For example, the function of AtNAP1;1 depends on its farnesylation motif—the wild-type AtNAP1;1 develops smaller leaves and cotyledons, while these organs are enlarged in plants expressing the mutant AtNAP1;1C369S (C369S transit at the CaaX motif) [57]. In our study, all HIPP proteins from *H. villosa* localized on PM, probably due their farnesylation, dependent on the CaaX motif. Most HMA-domain-containing proteins have heavy-metal-binding activity [23]; the membrane-associated characteristics of HIPPs-V may facilitate their binding affinities to specific heavy metals. Beside PM subcellular localization, several HIPPs-V also localized in other organelles, e.g., HIPP1/2/8/9/11-V had spot-like GFP signals in the cytoplasm; and except for HIPP2/5/9-V, all other HIPPs had nucleus GFP signals. Some HMA-domain-containing proteins participate in transporting heavy metals to maintain heavy-metal homeostasis [58]. HIPP1-V subcellularly localized both on PM and in cytoplasm. Its function in improving wheat tolerance to Cd may mediated by the binding and transportation of Cd for its homeostasis. NbHIPP26 is also subcellularly localized both on the plasma membrane and in the motile vesicles [35]. NbHIPP26, which plays roles in potato mop-top virus resistance, is related to the sensing and long-distance movement of viruses; it also regulates plant responses to drought stress [35]. The significance of the diverse subcellular localization of different HIPPs for their biological roles need to be further investigated.

## 4. Materials and Methods

### 4.1. Plant Materials

*H. villosa* (genome VV, accession no. 91C43) from the Cambridge Botanical Garden, Cambridge, U.K., was used for gene cloning and expression analysis. Commercial wheat variety Yangmai158, cultivated in the Institute of Agricultural Sciences in Lixiahe District of Jiangsu Province, was used as the transgenic receptor. *N. benthamiana* (NC89) plants were used for subcellular localization analysis. All plants were grown in a greenhouse and the growth conditions were as follows: 14/10 h day/night cycle, 24/20 °C day/night temperature, and 70% relative humidity.

### 4.2. Identification of the HIPP Gene Family in Triticeae Species

The *HIPP* genes from *Arabidopsis* and *Oryza* were based on those previously identified [23] and download from the National Center for Biotechnology Information (NCBI, https://www.ncbi.nlm.nih.gov/). The “HMA protein” in the annotated proteins database of *Hordeum vulgare* (HH, 2*n* = 2*x* = 14) [59] was searched and entries containing gene ID and protein sequences were obtained. We then performed a conserved-domains (CD) search (https://www.ncbi.nlm.nih.gov/Structure/cdd/wrpsb.cgi) and reserved the protein sequences that contained the HMA domain pfam00403.6 and the C-terminal isoprenylation motif CaaX. The presence of the isoprenylation motif, which was not characterized in the databases, was confirmed using PrePS [60].

The identified HIPP protein sequences of *H. vulgare* were used as query sequences to blast (*E*-value ≤ 10^−10^) against the protein databases of the other species, including *T. urartu* (AA, 2*n* = 2*x* = 14) [61], *T. aestivum* (AABBDD, 2*n* = 6*x* = 42) [62], *B. distachyon* (BdBd, 2*n* = 2*x* = 14) [63], *T. dicoccoides* (AABB, 2*n* = 4*x* = 28) [64], and *Ae. tauschii* (DD, 2*n* = 2*x* = 14) [65]. After removing the redundant gene sequences for each species, the alignment hits were validated by performing a CD search and PrePS as described above.

### 4.3. Cloning of HIPP Genes from Haynaldia villosa

Prediction of HIPP in *H. villosa* was carried out by comparing the transcriptomic data of *H. villosa* (data unpublished) with the HIPP protein sequences of *H. vulgare*, and the predicted proteins were processed according to the above method. According to the sequences of the *HIPP* genes from *H. vulgare*, the primers (Appendix A) were designed to clone the full-length cDNA of the *HIPP* genes from *H. villosa* using online software Primer3 (v. 0.4.0, University of California, Oakland, CA, USA). Mixed root, stem, and leaf tissue cDNA of *H. villosa* served as a template for the isolation. The specific primers for the *HIPP* genes from *H. villosa* were used for cloning. This was performed at 95 °C for 30 s, followed by 32 cycles of 95 °C for 30 s, 56 °C for 45 s, and 72 °C for 1 min, and then 5 min at 72 °C in Phanta Max Super-Fidelity DNA polymerase (Vazyme, Nanjing, China).

### 4.4. Subcellular Localization

Subcellular localization was performed as described by Zhao et al. [66] with the following modifications. The ORFs of the *HIPP-V* genes (without stop codons) were cloned into vector *pCambia1305-GFP* to produce the fusion expression construct 1305-GFP: HIPP-V. The plasma membrane (PM) marker construct (mCherry-SYP122) was provided by Prof. Yiqun Bao. The 1305-GFP fusion constructs and mCherry-SYP122 were transformed into *N. benthamiana* epidermal cells by *Agrobacterium tumefaciens* (strain GV3101) bacteria, incubated in darkness at 22 °C for 48–60 h, and the fluorescence signals were then observed by confocal microscopy (LSM780, Zeiss, Oberkochen, Germany).

### 4.5. Phylogenetic Analysis of the HIPP Gene Family

Phylogenetic analysis was performed as described by de Abreu-Neto et al. [23] with the following modifications. Multiple-sequence alignment was conducted using MUSCLE, which was integrated in MEGA X. The neighbor-joining analyses were performed in MEGA X using a p-distance parameter with 2000 bootstrap replications. Only the HMA domains were used to construct the phylogenetic tree.

### 4.6. Chromosome Distribution and Protein Structure Analysis of the HIPP Gene Family

Chromosomal information for the predicted *HIPP* genes from each species was obtained after using the cDNA sequences as a query sequence blasted to the genomic sequence to determine their chromosomal locations. Their locations were then drawn onto the physical map of each chromosome using Tbtools [67].

The domains of the wheat HIPP proteins were predicted using a CD search (https://www.ncbi.nlm.nih.gov/Structure/cdd/wrpsb.cgi). The corresponding evolutionary trees were constructed by MEGA X as described above. The formation of the protein structure was constructed using Tbtools [67].

### 4.7. RNA-seq Expression Analysis

The expression data were downloaded from expVIP to analyze the expression pattern of predicted wheat *HIPP* genes in different tissues (e.g., roots, leaves, spikes, and grains) under abiotic stress (e.g., drought and heat), and biotic stress (e.g., stripe rust and powdery mildew) [68], and the Z-score-normalized values of the TPM were used to construct the expression heat map using R version 3.5.1.

### 4.8. Transcription Abundance Analysis

*H. villosa* plants were grown in liquid Murashige and Skoog (MS) medium at 22–25 °C with a photoperiod of 12 h until the third-leaf stage. For expression analysis in response to Cd treatment, plants were inoculated with 1 mM CdCl_2_ and the leaves, stems, and roots were collected at 0, 1, 2, 6, 12, or 24 h after treatment. All of the samples were rapidly frozen in liquid nitrogen, then stored in an ultrafreezer (−80 °C) until use.

Total RNA was extracted using TRIZOL reagent according to the manufacturer’s protocol (Invitrogen, Carlsbad, CA, USA), and the sample was used to synthesize the first-strand cDNA using an RNA PCR Kit (TaKaRa, Shiga, Japan). Specific primers for *HIPP1-V* were used for qRT-PCR (Appendix A), and the *tubulin* gene was used as the internal control for normalization. qRT-PCR was performed with AceQ SYBR Green (Vazyme, Nanjing, China) using a Louts PCR 480 sequence detection system. The reactions were subjected to the following program: 95 °C for 1 min, 41 cycles of 95 °C for 10 s, and 60 °C for 30 s. The relative values of gene expression were derived from 2^−∆∆*C*T^ [69].

### 4.9. Genetic Transformation

*HIPP1-V* was cloned into the plant expression vector pBI220 (driven by the CAMV 35S promoter) to generate vectors pBI220: HIPP1-V. The method of stable transformation using bombardment was applied according to Xing et al. [70]. The vector pBI220: HIPP1-V, together with pAHC20 (carrying the *Bar* gene as a selectable marker), was co-transformed by bombardment into calli produced from young embryos of wheat cultivar Yangmai158. Regenerated plants were transplanted in the field and used for further characterization. Regenerated plants were analyzed via PCR using conjugated gene-specific primers CAMV35S-F and HIPP1-V-R (Appendix A) to identify positive transgenic plants.

### 4.10. Cd-Tolerance Analysis

For the Cd-tolerance tests, seeds of *HIPP1-V* transgenic plants and the receptor variety Yangmai158 were sterilized with 75% ethanol and 12% NaClO and washed five times with sterile water. To determine plant height and root length, the seeds were grown on 1/2 liquid Murashige and Skoog medium containing different concentrations of CdCl_2_ (e.g., 0, 5, 10, 20, 50, 100, 200, 500, 1000, 5000, 10,000 and 20,000 μM) for 15 days. The growth conditions were as follows: 14/10 h day/night cycle, 24/20 °C day/night temperature, and 70% relative humidity. The plant height and root length were measured and analyzed. The plant height and root length averages were computed based on the means of three independent experiments, each based on at least 10 independent plants, and the significance analysis was performed using Excel software.

## 5. Conclusions

The present study identified 278 *HIPP* genes from five *Triticeae* species and cloned 13 *HIPP* genes from *H. villosa*. Evolutionary and protein structure analyses showed that the *HIPP* genes are conserved in *Triticeae* species. The expression analysis of the *HIPP* gene family in common wheat and subcellular localization analysis of HIPPs-V showed that the functions of the *HIPP* genes may be involved in multiple pathways of plant development. *HIPP1-V* was upregulated in the roots, stems, and leaves of *H. villosa* in response to Cd treatment, and overexpression of *HIPP1-V* enhanced the tolerance of common wheat to Cd. Our study systematically analyzed the function and evolution of the *HIPP* gene family in *Triticeae* species and demonstrated the important role of *HIPP1-V* in heavy-metal homeostasis and detoxification.

## Figures and Tables

**Figure 1 ijms-21-06191-f001:**
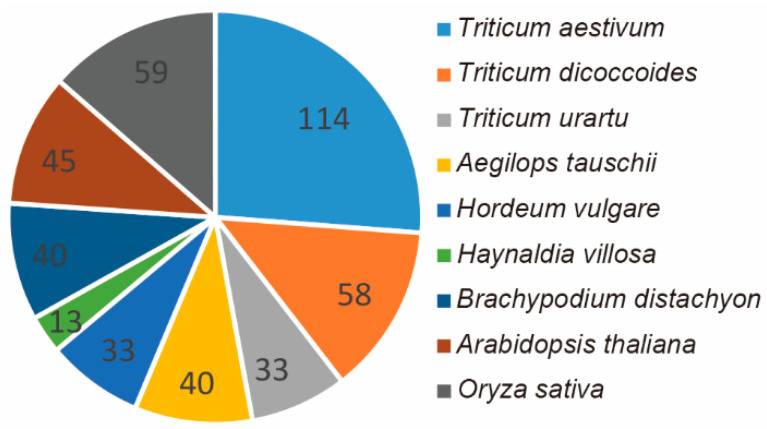
The total number of heavy-metal-associated isoprenylated plant protein (*HIPP*) genes in the nine species of *T. aestivum*, *T. dicoccoides*, *T. urartu*, *Ae. tauschii*, *H. vulgare*, *H. villosa*, *B. distachyon*, *A. thaliana,* and *Oryza sativa*.

**Figure 2 ijms-21-06191-f002:**
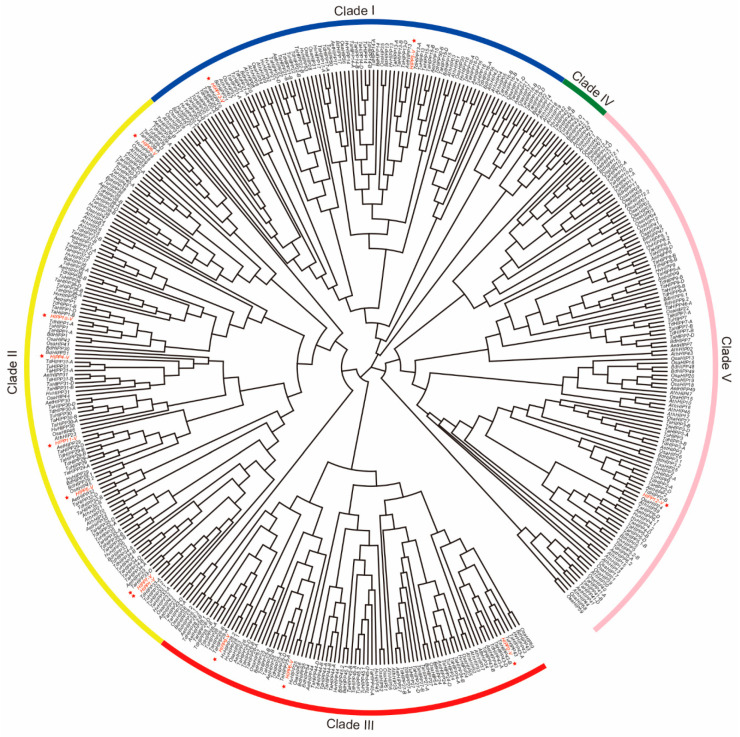
The phylogenetic analysis of the *HIPP* gene family from nine plant species. Species abbreviations: Ta: *T. aestivum*; Tu: *T. urartu*; Aet: *Ae. tauschii*; Td: *T. dicoccoides*; Hv: *H. vulgare*; Bd: *B. distachyon*; Ath: *A. thaliana*; Osa: *O. sativa*; -V: *H. villosa*. The red asterisks indicate the *HIPP* genes from *H. villosa*.

**Figure 3 ijms-21-06191-f003:**
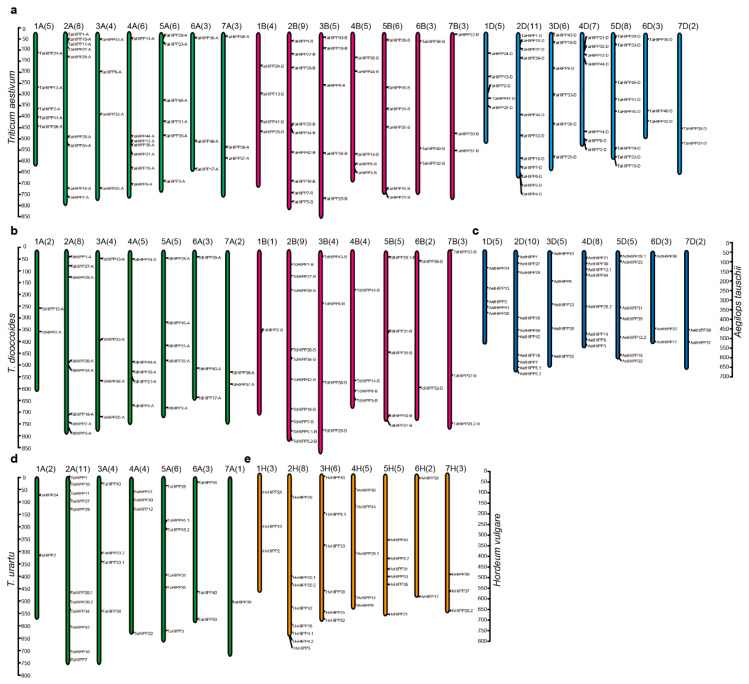
Chromosomal distribution of the *HIPPs* in the five *Triticeae* species. The chromosomal distribution of the *HIPP* genes in *T. aestivum* (**a**), *T. dicoccoides* (**b**), *Ae. tauschii* (**c**), *T. urartu* (**d**), and *H. vulgare* (**e**). The identities of the chromosomes are indicated at the top of each chromosome—the number in brackets corresponds to the number of genes located on the corresponding chromosome, while the *HIPP* gene names are shown to the right of each chromosome. Ta: *T. aestivum*; Tu: *T. urartu*; Aet: *Ae. tauschii*; Td: *T. dicoccoides*; Hv: *H. vulgare*.

**Figure 4 ijms-21-06191-f004:**
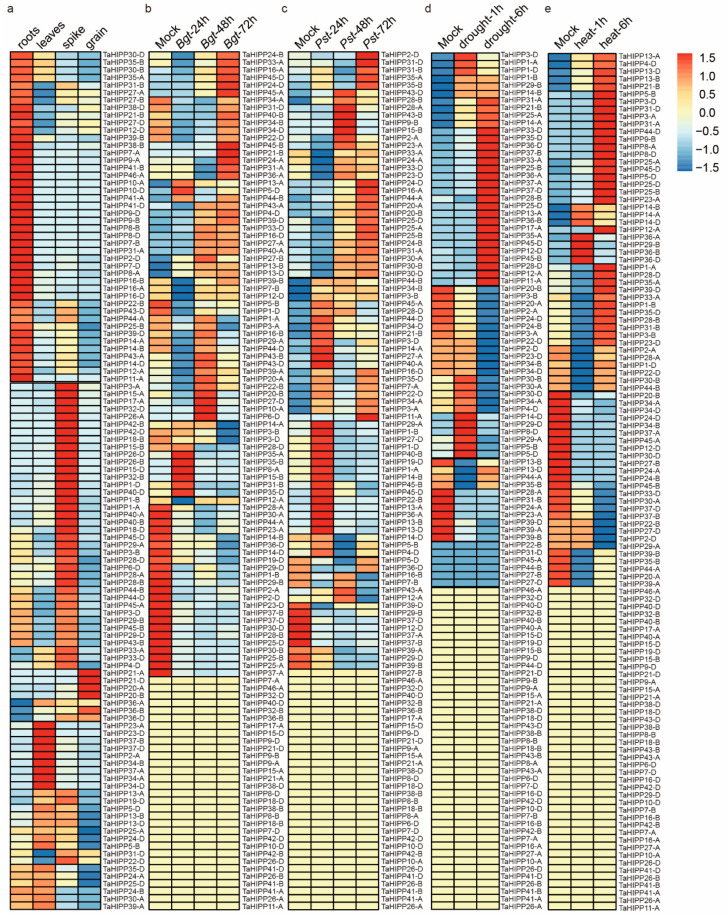
Heat map of the expression profiling of common wheat *HIPPs* in different tissues and under various stresses: the expression of *HIPPs* in different tissues (**a**), and their response to powdery mildew (**b**), yellow rust (**c**), drought (**d**), and heat treatment (**e**). The expression signal of each gene was based on the Z-score-normalization value. Abbreviations: *Bgt*: powdery mildew; *Pst*: yellow rust.

**Figure 5 ijms-21-06191-f005:**
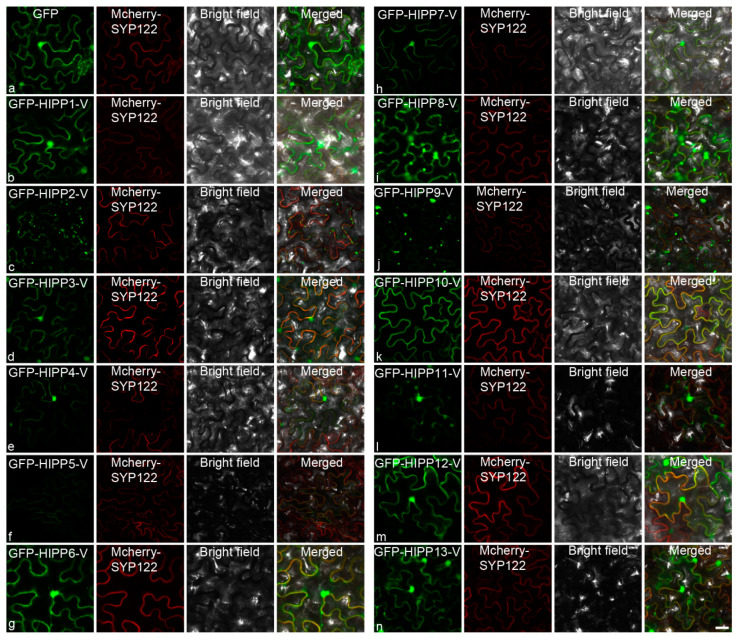
Subcellular localization of HIPPs in the epidermal cells of *Nicotiana benthamiana*: subcellular localization of GFP (**a**), GFP-HIPP1-V (**b**), GFP-HIPP2-V (**c**), GFP-HIPP3-V (**d**), GFP-HIPP4-V (**e**), GFP-HIPP5-V (**f**), GFP-HIPP6-V (**g**), GFP-HIPP7-V (**h**), GFP-HIPP8-V (**i**), GFP-HIPP9-V (**j**), GFP-HIPP10-V (**k**), GFP-HIPP11-V (**l**), GFP-HIPP12-V (**m**), and GFP-HIPP13-V (**n**). GFP was used as the control. The localization of mCherry-SYP122 is shown in red, and the localization of GFP and its fusion proteins are shown in green. Scale bar = 10 µm.

**Figure 6 ijms-21-06191-f006:**
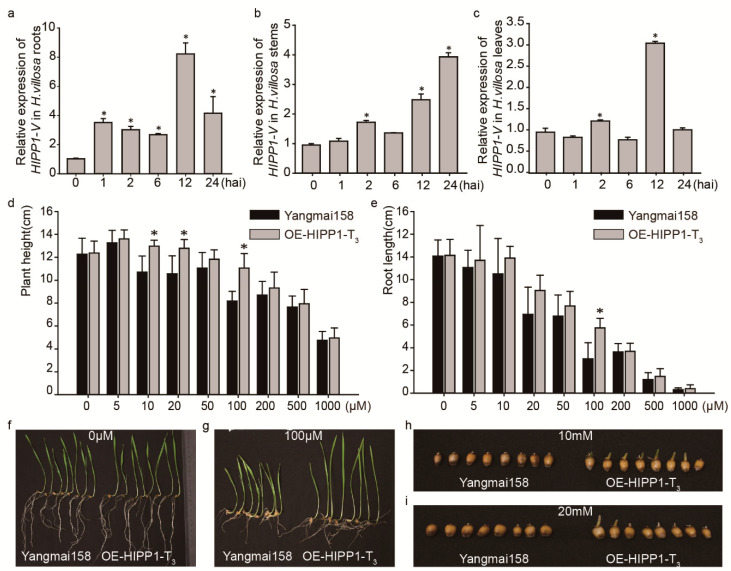
Tolerance of *HIPP1-V* transgenic plants to Cd. Time-course expression profiling of *HIPP1-V* of *H*. *villosa* in response to Cd treatment in the roots (**a**), stems (**b**), and leaves (**c**). (**d**) The plant height between OE-HIPP1-T_3_ and Yangmai158 plants exposed to different Cd concentrations. (**e**) The root length between OE-HIPP1-T_3_ and Yangmai158 plants in different Cd concentrations. (**f**–**i**) OE-HIPP1-T_3_ and Yangmai158 phenotypes under treatments of different Cd concentrations (0, 100 μM, 10 mM, and 20 mM). **p* < 0.05. hai: hours after induction.

**Table 1 ijms-21-06191-t001:** Number of *HIPPs* from different species on each of the chromosomes.

Chromosome	*T. aestivum*			*T. dicoccoides*		*T. urartu*	*Ae. tauschii*	*H. vulgare*	Total
	A	B	D	A	B	A	D	H	
1	5	4	5	2	1	2	5	3	27
2	9	9	11	8	10	11	10	8	76
3	4	5	6	4	4	4	5	6	38
4	7	5	7	5	4	4	8	5	45
5	6	6	8	5	5	6	7	5	48
6	3	3	3	3	2	3	3	3	23
7	3	3	2	2	3	1	2	3	19
Unknown *						2			2
Total	37	35	42	29	29	33	40	33	278

***** These genes were assigned to an unknown chromosome.

**Table 2 ijms-21-06191-t002:** Information of the *HIPP* gene family in *H. villosa.*

Name	Homologs in *H. vulgare*	ORF	aa	Location of HMA Domain (aa)	No. of HMA Domain	PI	MW (kDa)
*HIPP1-V*	*HvHIPP33*	456	151	37–91	1	9.61	16.63
*HIPP2-V*	*HvHIPP43*	1251	416	16–63	1	7.54	44.21
*HIPP3-V*	*HvHIPP13*	1023	340	22–76, 162–204	2	9.63	36.24
*HIPP4-V*	*HvHIPP31*	486	161	35–94	1	10.01	17.81
*HIPP5-V*	*HvHIPP31*	489	162	36–95	1	10.3	17.64
*HIPP6-V*	*HvHIPP25*	999	332	12–73	1	5.62	36.36
*HIPP7-V*	*HvHIPP33*	456	151	37–91	1	9.78	16.59
*HIPP8-V*	*HvHIPP39*	507	168	10–61	1	8.41	18.72
*HIPP9-V*	*HvHIPP28.1*	921	306	12–59	1	4.86	33.35
*HIPP10-V*	*HvHIPP36*	468	155	35–90	1	10.04	17.35
*HIPP11-V*	*HvHIPP29*	489	162	39–98	1	9.97	17.64
*HIPP12-V*	*HvHIPP2*	498	165	11–60	1	6.79	17.92
*HIPP13-V*	*HvHIPP20.2*	1065	354	71–122, 167–227	2	5.21	39.23

Note: ORF: open reading frame; aa: amino acids; HMA: heavy-metal-associated; PI: isoelectric point; MW: molecular weight.

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
