# Peer review of "Characterization of the Heavy-Metal-Associated Isoprenylated Plant Protein (HIPP) Gene Family from Triticeae Species"

_ijms, 2020, doi:10.3390/ijms21176191_

Round 1

Reviewer 1 Report

In this paper, the author started this study from the following perspective " HIPPs are a group of metal-binding metallochaperones, which contained HMA domain and a C-terminal isoprenylation  CaaX motif.” In particular, it is significant in that the characteristics of the genes of HIPPs were identified through genome-wide analysis. However, it is difficult to draw the conclusion that “HIPP gene family presents functional differentiation during the evolution in common wheat”. Functional researches of HIPPs gene family members have not been conducted, even if some cd resistances were analyzed. Most of the main conclusions were obtained through domain analysis. In addition, it is considered that the confocal data observed after transient transformation of GFP fusion constructs leads to differences in the function of gene members to the intracellular location. Therefore, if you want to highlight the meaning of this paper, it would be better to focus on the genome-wide analysis section and write the paper intensively about those points. It seems unreasonable to draw conclusions from domain analysis. Since most of the functional explanations of gene members are conclusions based on existing literature, the conclusions drawn through this study are emphasized too much. I think it will be necessary to correct the native speakers of English.

Author Response

1. In this paper, the author started this study from the following perspective "HIPPs are a group of metal-binding metallochaperones, which contained HMA domain and a C-terminal isoprenylation CaaX motif.” In particular, it is significant in that the characteristics of the genes of HIPPs were identified through genome-wide analysis.

Reply: Thank you.

2. However, it is difficult to draw the conclusion that “HIPP gene family presents functional differentiation during the evolution in common wheat”. Functional researches of HIPPs gene family members have not been conducted, even if some cd resistances were analyzed. Most of the main conclusions were obtained through domain analysis.

Reply: We agree we did not perform intensive functional analysis of each gene member, and it is not sufficient to draw conclusion for their differentiation only based on the domain or subcellular differences, as well as expression analysis in silico. Thus,we have deleted this point of conclusion associated with “HIPP functional differentiation” throughout the manuscript. Instead, we only speculate that HIPPs might regulate different biological processes concerning plant development and various response to stress through the surveys of gene expression, subcellular localization and etc.

3. In addition, it is considered that the confocal data observed after transient transformation of GFP fusion constructs leads to differences in the function of gene members to the intracellular location. Therefore, if you want to highlight the meaning of this paper, it would be better to focus on the genome-wide analysis section and write the paper intensively about those points. It seems unreasonable to draw conclusions from domain analysis.

Reply: This is a very nice comment. We agree to focus on the genome-wide analysis of this gene family in this paper. We rewrote this part and only highlighted the differences in subcellular localization of different HIPP members.

“The domain analysis” under “discussion 3.2” in the original manuscript was also deleted in the revised manuscript.

4. Since most of the functional explanations of gene members are conclusions based on existing literature, the conclusions drawn through this study are emphasized too much.

Reply: Yes, determine of gene function need more intensive studies. We revised the manuscript in the “Discussion” parts, we only gave the information of the function of some HIPPs which have been intensive studied. The function of other HIPPs need to be studied further in future.

5. I think it will be necessary to correct the native speakers of English.

Reply: We have asked MDPI English pre-edit services to proofread the manuscript.

Reviewer 2 Report

The authors presented very interesting study related to the heavy metal-associated (HMA) isoprenylated plant proteins in Triticeae species combining the in silico expression profiling and experimental observation using transgenic wheat plants overexpressing HIPP1 in high cadmium content. The study is, definitely, original and novel, with a high practical relevance.

The manuscript is very well introduced, stating clear aims and focus of the study. The methods are presented in appropriate way. The presentation of the results is standard and good. The results are appropriately discussed, but some revision is needed (see comments below). 

Overall, the manuscript is valuable and interesting, and I believe it can be attractive for the readers.

I recommend to accept the manuscript after addressing my comments below within minor revision.

Comments:

  1. The study is well written, but the language need to be improved. I recommend the correction of the manuscript by the English native speaker.
  2. The experiment with cadmium is insufficiently described. For example, number of plants per treatment or number of repeated experiments should be specified.
  3. The statistical analysis should be clearly specified.

Author Response

1. The study is well written, but the language need to be improved. I recommend the correction of the manuscript by the English native speaker.

Reply: We have asked MDPI English pre-edit services to proofread the manuscript.

2. The experiment with cadmium is insufficiently described. For example, number of plants per treatment or number of repeated experiments should be specified.

Reply: Sorry for our insufficient description of the methods for some experiment. We have added details on this experiment design in “4.10. Cd tolerance analysis” under “4. Materials and Methods”.

3. The statistical analysis should be clearly specified.

Reply: We have described the statistical analysis method in “Materials and Methods” section 4.10 and in legend of Figure 6.